# Visual attention prediction improves performance of autonomous drone racing agents

**Christian Pfeiffer**[1,2]*, **Simon Wengeler**[1,2], **Antonio Loquercio**[1,2], **Davide Scaramuzza**[1,2]

**1** Robotics and Perception Group, Department of Informatics, University of Zurich, Zurich, Switzerland,
**2** Department of Neuroinformatics, University of Zurich and ETH Zurich, Zurich, Switzerland

* christian.pfeiffer@uzh.ch

**Data Availability Statement:** The dataset used in this study is available in an Open Science Framework repository (https://osf.io/uabx4/, Dataset DOI: 10.17605/OSF.IO/UABX4).

## Abstract

Humans race drones faster than neural networks trained for end-to-end autonomous flight. This may be related to the ability of human pilots to select task-relevant visual information effectively. This work investigates whether neural networks capable of imitating human eye gaze behavior and attention can improve neural networks' performance for the challenging task of vision-based autonomous drone racing. We hypothesize that gaze-based attention prediction can be an efficient mechanism for visual information selection and decision making in a simulator-based drone racing task. We test this hypothesis using eye gaze and flight trajectory data from 18 human drone pilots to train a visual attention prediction model. We then use this visual attention prediction model to train an end-to-end controller for vision-based autonomous drone racing using imitation learning. We compare the drone racing performance of the attention-prediction controller to those using raw image inputs and image-based abstractions (i.e., feature tracks). Comparing success rates for completing a challenging race track by autonomous flight, our results show that the attention-prediction based controller (88% success rate) outperforms the RGB-image (61% success rate) and feature-tracks (55% success rate) controller baselines. Furthermore, visual attention-prediction and feature-track based models showed better generalization performance than image-based models when evaluated on hold-out reference trajectories. Our results demonstrate that human visual attention prediction improves the performance of autonomous vision-based drone racing agents and provides an essential step towards vision-based, fast, and agile autonomous flight that eventually can reach and even exceed human performances.

## Introduction

First-person view (FPV) drone racing is an increasingly popular televised sport in which human pilots compete to complete challenging obstacle courses in a minimum time. Using only visual feedback from an FPV camera attached to the teleoperated unmanned aerial vehicle, human pilots are able to plan and execute appropriate control actions to navigate the drone along challenging race tracks [1, 2]. The visual-motor coordination skills required to

**Funding:** This work was supported by the Ernst Göhner Foundation and University of Zurich Alumni Fonds zur Förderung des Akademischen Nachwuchses (FAN Fellowship), by the National Centre of Competence in Research (NCCR) Robotics through the Swiss National Science Foundation (SNSF) and the European Union's Horizon 2020 Research and Innovation Programme under grant agreement No. 871479 (AERIAL-CORE) and the European Research Council (ERC) under grant agreement No. 864042 (AGILEFLIGHT). The funders had no role in study design, data collection and analysis, decision to publish, or preparation of the manuscript.

**Competing interests:** The authors have declared that no competing interests exist.

achieve top-level performances in drone racing are based on many years of repeated practice and flight experience in drone racing simulators and real-world races [2, 3]. But, how exactly is visual perception related to aircraft control? Recent experimental evidence indicates a strong relationship between human drone racing pilots' eye gaze behavior and future flight trajectories and shows that the direction of eye gaze fixation precedes planned control actions [2]. Thus, visual attention measured by eye gaze fixations indicates a human pilot's intention and subsequent control action. Because quadrotor drones are extremely agile vehicles, they become increasingly relevant in time-critical missions, such as search and rescue, aerial delivery, and industrial inspection tasks. Therefore, over the last decade, research on autonomous, agile quadrotor flight has pushed platforms to higher speeds and agility [4–12] In this line of research a key question is: Can we design an algorithm for fully autonomous vision-based fast and agile drone flight that performs as well as or better than human pilots? Solving this challenge is one of the most pertinent goals in autonomous vision-based quadrotor navigation, reflected in an increasing number of simulation-based [13, 14] and real-world competitions [15, 16]. The challenges are enormous, particularly regarding the issues of low-latency perception-aware planning and state estimation under motion blur [16]. If solved, numerous benefits outside of drone racing would arise. This includes low-latency agile autonomous systems that perform safe and effective missions in unknown, cluttered environments inaccessible to humans for industrial inspection and search and rescue applications. The two leading approaches are model-based and learning-based system design. The model-based approach follows a classical sense-plan-control scheme, which is modular, and requires very accurate knowledge about the drone dynamics, the drone's state, and the ability to perform low-latency minimum-time control onboard [8, 12, 15]. Indeed, this approach has been very successful and has been able to outperform experienced drone racing pilots on challenging race maneuvers in highly controlled environments [8]. However, model-based approaches often require external sensing and highly accurate systems knowledge, pre-planned trajectories, and do not generalize to unknown environments or noisy sensory inputs. The alternative is infusing learning-based methods into systems design, where sensing, planning, and control tasks are performed by a single neural network. These so-called end-to-end neural networks have been successfully trained and deployed for quadrotor flights of acrobatic maneuvers [11], obstacle avoidance in the wild [17], and simulator-based drone racing [18, 19]. Surprisingly, none of these previous works have considered imitating or making use of flight trajectories and visual-motor coordination behavior produced by experienced human drone racing pilots. The main objective of this work is to answer the question of whether gaze-based visual attention prediction can improve the performance of end-to-end models for vision-based autonomous drone racing beyond state-of-the-art. We address the problem of a lack of human ground truth data during deployment by training a neural network for predicting human visual attention from RGB images. The scope of the present work is an evaluation of the flight performances of end-to-end controller architectures for the task of vision-based autonomous drone racing in a highly realistic simulator.

## Contributions

The main contributions of this work are:

1. We train and evaluate a visual attention prediction model for autonomous drone racing.

2. We train end-to-end deep learning networks using imitation learning that can complete a challenging race in a vision-based drone racing task, with a performance as good as human pilots.

3. We demonstrate that attention prediction models outperform models using raw image inputs and image-based abstractions (i.e., feature tracks).

4. We found a better generalization performance to previously unseen flight trajectories for end-to-end drone racing agents using attention prediction or feature tracks when compared to a raw image input baseline.

The Related Work section describes related works in the domain. The Materials and Methods section describes the datasets, network architectures, and experimental analysis methods used in this work. The Results section presents experimental results obtained for the visual attention prediction, control command prediction, and end-to-end drone racing performance. The Discussion section relates the experimental findings to previous work and proposed future work. The Conclusion section concludes the paper.

## Related work

Behavioral cloning, or imitation learning, has the goal to develop neural networks that can map from sensory inputs to control actions by learning from (human) expert data in a supervised fashion [20, 21]. The main benefit of imitation learning is that it does not require feature engineering. Imitation learning approaches were initially developed and successfully deployed for car driving applications, such as lane following and obstacle avoidance [19, 22]. A caveat however is that training models on expert data often do not provide information about the states that deviate from the experts, which can lead to failure if the agent encounters such states. This can be mitigated by dataset aggregation (DAgger), where novel training data is collected while training a primary policy on a reference policy [23] or by introducing displacements [18] or distortions to control commands [19] to enlarge the state space for training. Dataset aggregation has been successfully used for training end-to-end networks for autonomous car driving [19] and autonomous quadrotor flight [11]. Another shortcoming of imitation learning is that it does not allow the network to compensate for mistakes made by the expert. A possible solution is the use of observational imitation learning in which a network learns to select optimal behavior while observing multiple imperfect teachers. This approach outperformed reinforcement learning and imitation learning approaches in vision-based autonomous drone racing in a simulator [24]. However, not only the choice of network architecture and training method but also the choice of input/output representation strongly affect network performance. Abstractions of either input or output data typically outperform networks operating directly on raw image data. For instance [11], observed better performance in autonomous acrobatic flight using feature tracks than using RGB images directly. Similarly [25], found better 3D localization performance using grayscale instead of RGB images. Likewise [26], found better performances in autonomous car racing when predicting parameterized trajectories for a model predictive controller (MPC) driving the car compared to letting the network predict control commands directly. Such sensory and output abstractions seem advantageous in network performance and generalization ability. It should also be noted that several previous works follow hybrid approaches combining learning methods for perception [27] and localization [28] with model-based methods for planning [29] and control [21] and have demonstrated successes. However, these approaches often require extensive system identification and controller tuning, which are not required when using end-to-end neural network controllers. In this study, we investigate whether imitating human visual attention and flight behavior, could serve to improve the performance of state-of-the-art end-to-end models on autonomous drone racing tasks, which requires the models to perform fast and agile flight through mandatory waypoints (i.e., race gates). The importance of visual attention in vision-

based navigation has not only been demonstrated in drone pilots [2]. Human car drivers move their eye gaze to future waypoints and driving paths several seconds and meters ahead of the current position of the car [30]. These eye gaze fixations allow the operator to compensate for unwanted visual image motion (retinal stabilization) and estimate the current vehicle motion. Most importantly, there is a strong temporal and spatial relationship between eye gaze fixations and subsequent control commands. Drivers execute control actions congruent with the eye gaze deviations from the vehicle's forward velocity at fixed temporal offsets of 400 ms for driving on winding roads [31]. Gaze monitoring in car drivers also provides valuable information for autonomous driving agents, in particular regarding high-level intentions, such as whether to perform a left or right turn [32]. It can even support more efficient performance by selecting only task-relevant information [33]. Previous works have tried to extract information from eye gaze for steering cars, e.g., for assistive technology, hands-free operation [34], attention or intention monitoring [35], or for teaching autonomous agents to drive in virtual cities [33]. However, those applications are usually slow, use limited control commands, and have not directly used visual attention for fast and agile drone flight.

## Materials and methods

### Ethics statement

The study protocol was approved by the local Ethical Committee of the University of Zurich and the study was conducted in line with the Declaration of Helsinki. All participants gave their written informed consent before participating in the study. All human data taken from a publicly available dataset were fully anonymized before we accessed them.

### Human drone racing dataset

We use the publicly available "Eye Gaze Drone Racing Dataset" (Open Science Framework repository: https://osf.io/gvdse/), originally released by [2], which consists of eye gaze, control commands, drone state ground-truth, and the FPV video (800 × 600 pixels resolution) recordings from experienced drone pilots flying in a drone racing simulator (Fig 1a–1c illustrates the experiment setup). The eye gaze data is projected onto the screen to obtain gaze locations that correspond with the recorded videos. For this study, we randomly select flight trajectory data from 36 collision-free flights from 18 human pilots from a figure-eight race track (see example trajectory in Fig 1d). Flight trajectory selection is constrained by the achieved lap time, that is we randomly select data within one interquartile range of the group median lap time (11.80 sec) and assign these data randomly to the training set (18 trajectories; median lap time = 11.69 sec, min = 10.79 sec, max = 14.46 sec) and test set (18 trajectories; median lap time = 11.83 sec, min = 11.05 sec, max = 14.91 sec; paired-samples t-test shows no statistical difference in lap times between training and test set).

Because the AlphaPilot drone racing simulator used for drone state data logging by [2] is proprietary software that did not allow for closed-loop control, we use the open-source drone racing simulator Flightmare [36], which is tailored to machine learning tasks as required for the present study. The quadrotor platform had an arm length of 17 cm, an all-up-weight of 1 kg, a maximum collective thrust of 21.7 N, and a maximum rotational velocity of 6 rad/s. The RGB camera had a horizontal field-of-view of 80˚, and an uptilt angle of 25˚. We thus used the ground-truth trajectory, eye gaze, drone, and camera settings of the original dataset by [2] to generate a novel ground-truth dataset required for network training and evaluation. We designed a visual environment largely identical in color and dimensions, with identical gate sizes, positions, and shapes as used by [2]. We then rendered the drone ground-truth poses in Flightmare to collect images of the same resolution as in the original dataset, which is

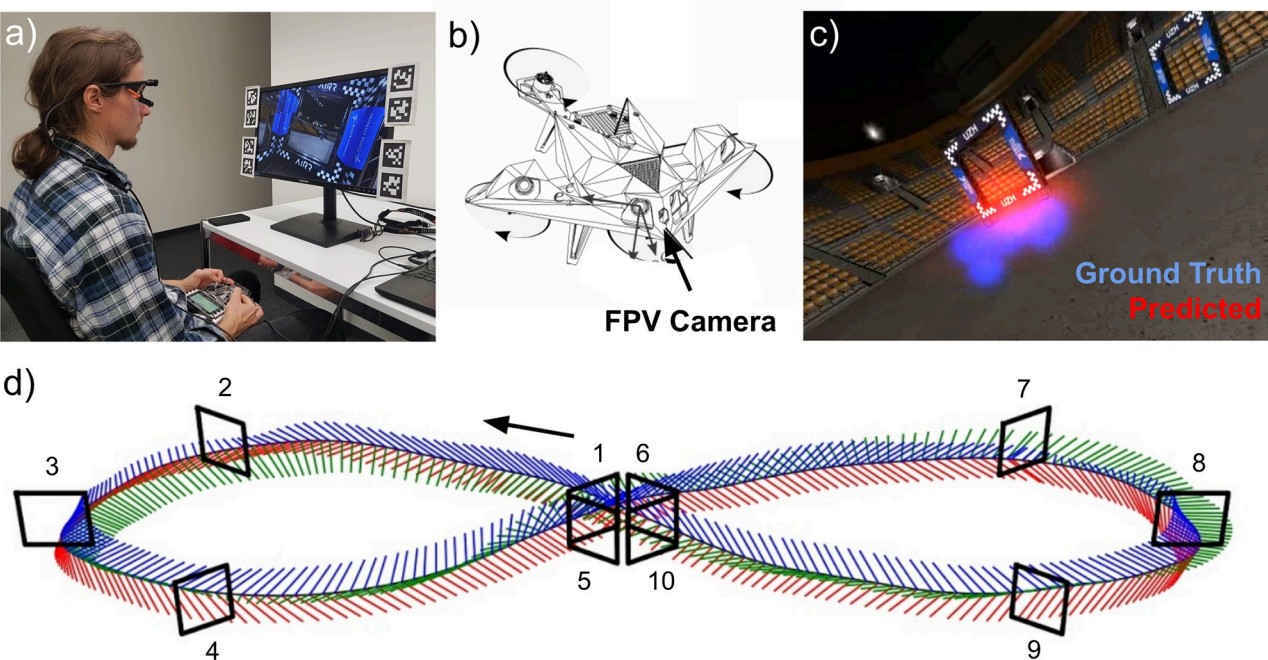

**Fig 1. Experimental methods illustrated.** a) Experimental setup used in [2]; b) First-person view (FPV) racing drone; c) Example FPV image showing racing gates, gaze-based, and network-predicted attention maps. d) Reference trajectory by a human pilot showing quadrotor axes in red (x), green (y), and blue (z). Race gates are represented by black rectangles and numbered in sequence. Black arrow indicates the direction of flight.

subsequently used for attention network training. Although gaze fixations can be used to indicate the pilots' focus of attention, the uncertainty inherent in the measurements can better be expressed using a probability distribution over the image coordinates. Using the procedure described in [37], we generate ground-truth continuous visual attention maps $A_t$ by averaging the gaze positions recorded for each frame (in pixels) and using these fixations $f_t$ from the frame at time $t - 12$ to $t + 12$ (a total of 25 frames at 60Hz) to define a 2D multivariate Gaussian distribution (with a fixed diagonal variance matrix $\Sigma = \text{diag}(200, 200)$) centered on each fixation. For each pixel, the maximum value across these Gaussians is computed to create a visual attention map over the image:

$$A_t(x, y) = \max_{i \in \{t-12,...,t+12\}} \mathcal{N}(x, y; f_i, \Sigma).$$

(1)

To form a valid probability distribution of the pilot's visual attention, this attention map is normalized to sum to one. An example of one of these ground-truth attention maps can be seen as the output of the architecture shown in Fig 2. We filter out any laps with crashes or in which the drone does not pass through all gates, and also perform a manual inspection of the trajectories, removing those that are undesirable for training a controller, e.g. when pilots considerably deviate from the figure-eight reference trajectory (Fig 1c). Furthermore, we only use frames where both gaze and control ground truth is available. This results in a total of 675, 251 valid frames from 18 subjects. The gaze dataset is split into a training set with 508, 670 frames and a test set with 166, 581 frames, with both sets containing samples from all included subjects but not from the same individual experimental runs. This dataset is used for the training and performance evaluation of the visual attention prediction network. The dataset used in

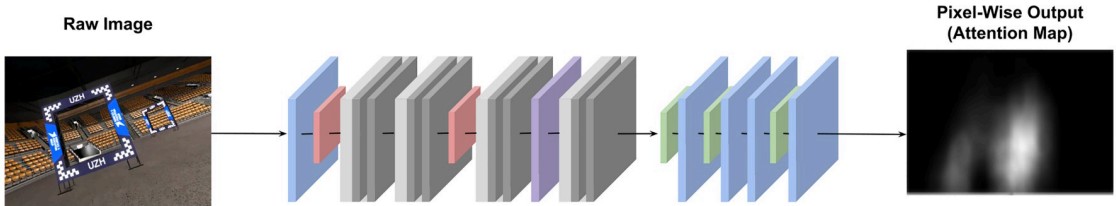

**Raw Image**

**Pixel-Wise Output
(Attention Map)**

**Fig 2. The architecture of our attention-prediction network based on ResNet-18.** The network predicts pixel-wise attention probabilities and is therefore a Fully Convolutional Network. ResNet blocks (each with two convolutional layers) are shown in grey, convolutional layers in purple and blue (with and without batch normalization), max-pooling layers in red and upsampling layers in green.

this study is available in an Open Science Framework repository (https://osf.io/uabx4/, Dataset DOI: 10.17605/OSF.IO/UABX4).

## Visual attention prediction network

Fig 2 pictures the architecture of the visual attention prediction network, based on [38], which is designed to predict visual attention as a distribution over image pixels. The network uses ResNet-18 [39] layers pre-trained on ImageNet [40] and is trained on individual frames. It uses the first four residual blocks of the ResNet-18 architecture, including strided convolution and pooling operations. To maintain a high spatial resolution for predicting attention maps, the model is trained on RGB images of size $400 \times 300$ (half the original resolution), resulting in feature maps of resolution $25 \times 19$ after being processed by the encoder. These features are repeatedly upsampled and passed through convolutional layers with ReLU activations, finally obtaining a visual attention map of the same resolution as the input image by applying a 2D softmax to create a valid probability distribution. Similar to [37], Kullback-Leibler divergence is used to compute the loss:

$$D_{KL}(A \parallel \hat{A}) = \sum_{x,y} A(x,y) \log\left(\frac{A(x,y)}{\hat{A}(x,y)}\right) \qquad (2)$$

where $A$ is the ground-truth attention distribution, $\hat{A}$ is the network's prediction, and $x$ and $y$ are image coordinates. The visual attention prediction network is trained for 5 epochs with a batch size of 128 and using the Adam optimizer [41] with a learning rate of $2 \times 10^{-4}$. During training, we use data augmentation by randomly applying the following transformations to the input images: brightness, contrast, saturation and hue changes, the addition of Gaussian noise, applying Gaussian blur, and erasing of random image regions. The trained network is ultimately used to obtain encoder features as input to the end-to-end drone racing agent.

## End-to-end controller network

Fig 3 shows the architecture of the visual attention-prediction based end-to-end drone racing network. The architecture is adapted from the "Deep Drone Acrobatics" (DDA) architecture proposed in [11]. It takes as input a short history of measurements: reference states in world coordinates consisting of rotation, linear and angular velocity (sampled from the reference trajectory at 50 Hz), and a state estimate, also entailing rotation, linear and angular velocity (sampled at 100 Hz). Note that unlike in [11], we do not use the original implementation in ROS designed for real-world quadrotor flight but instead use a custom Python 3.8 implementation

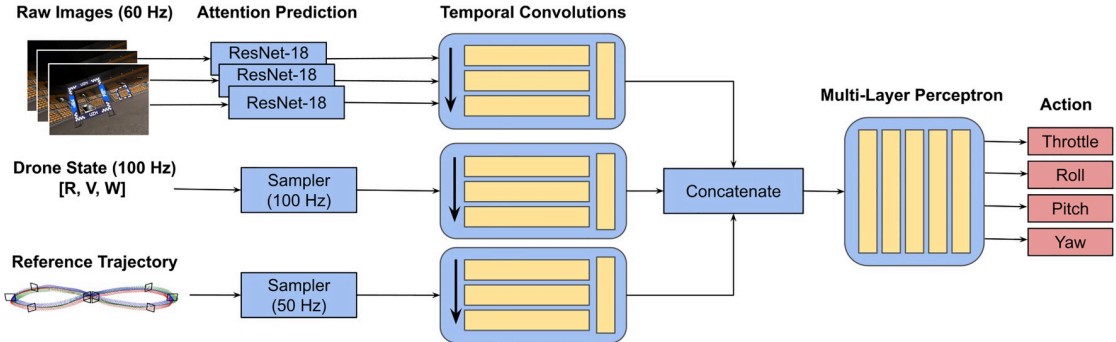

**Fig 3. Architecture of the attention-prediction based end-to-end controller.**

of the code compatible with the Flightmare simulation environment. Moreover, we use ground-truth states as a substitute for state estimates. The inputs for each of the described branches are processed by temporal convolutions before being concatenated and passed through the control module consisting of four linear layers and predicting mass-normalized thrust and body rates. We introduce one major modification to the original network architecture by replacing feature tracks with encoder features from visual attention prediction as an input to the network. We flatten the features extracted by the encoder of the visual attention network (i.e., $25 \times 19$ features) to a one-dimensional vector of size 475 at each time step. These vectors are then further processed by temporal convolutions like the other inputs. The control module is identical to the original network architecture used by [11]. For performance comparison, we use two baseline models, which are identical to the visual attention-prediction based model, apart from the visual attention input. The first baseline model is an end-to-end drone racing network receiving raw RGB images as inputs (i.e., $400 \times 300 \times 3$ features), which are stacked in the feature dimension and processed by a 2D convolutional network before also being transformed to a single vector as input to the control module. The second network is an end-to-end drone racing network receiving feature tracks as inputs. Feature tracks are an abstraction of visual inputs, initially used in [11] to provide a better transfer from learning in simulation to control in the real world. We use a re-implementation of feature tracks from the VINS-Mono package [42] in Python. Feature tracks are represented as a five-dimensional vector: the location of salient image features in normalized image coordinates, the velocity of features tracked over subsequent frames, and the number of time steps each feature has been tracked. Features are extracted using the Harris corner detector [43] and tracked using the Lucas-Kanade method [44]. Outliers are removed using geometric verification and key point correspondences of more than one pixel from the epipolar line. Exactly 40 feature tracks per time step are used as input to the respective controller (i.e., $40 \times 5$ features), sampled from all tracked features. The feature-track based controller receives feature tracks after they are passed through a reduced version of the PointNet architecture [45]) as input to the temporal convolution part of the network.

## End-to-end controller training

We use the same training strategy employed in [11], using imitation learning with DAgger [46]. We train each model on 18 reference trajectories of the training data. Using these human-generated trajectories ensures that the quadrotor's camera is pointed in the direction of movement, and meaningful attention predictions can be made based on models trained on

human gaze data. An MPC expert with access to the ground-truth state is used that follows the trajectory, providing labels for network predictions. It uses the simplified quadrotor model proposed in [47]. It solves the optimization problem of minimizing the difference between the reference trajectory and the predicted quadrotor states, subject to the quadrotor dynamics (see [11] for more details). Exploration—and thus larger coverage of the state-space—is facilitated by adding random noise to the expert command with a small probability, which increases throughout data generation and network training. Additionally, the network predictions (rather than the expert predictions) are executed if they are within a boundary close to the expert command, the range for which also increases over time. We record data for 30 rollouts before training for 20 epochs, which is repeated five times for a total of 150 rollouts and 100 epochs of training.

### Drone racing performance evaluation

We evaluate end-to-end drone racing network performances on 18 reference trajectories of the training set by comparing the performances between visual attention prediction, raw RGB images, and feature track-based networks. To evaluate network generalization, we evaluate network performance on hold-out test set trajectories that the networks have not observed previously. For each scenario, we perform 10 repetitions of the test flight to compute the number of gates successfully passed. This metric is computed considering the period between the start of the network-controlled flight until completing the trajectory or until collision with a gate, the ground floor, or virtual collider boundaries placed at $30 \times 15 \times 8$ meters around the racing track.

## Results

In this study, we trained two kinds of neural networks: one that predicts human gaze-based visual attention from RGB images (attention prediction model) and one that uses attention prediction to control a racing drone in a vision-based autonomous drone racing task (attention-based end-to-end controller). The following sections present a performance evaluation of the visual attention prediction model, the control command prediction performance of the end-to-end controller, the drone racing performance on seen trajectories (training set), and the generalization performance to hold-out trajectories (test set).

### Visual attention prediction performance

Fig 4 provides a qualitative assessment of the predictions of the visual attention prediction model on exemplar images. When gates are in clear view of the FPV camera (as compared to, e.g., the moment of traversal), attention predictions match ground-truth data very well both in terms of location and accumulating probability mass in one region. This also holds when multiple gates are in view. In these cases, the network's predictions mostly focus on the upcoming gate, just like the human ground-truth [2].

We evaluate visual attention prediction performance by comparison to two simple baselines. The first consists of the mean attention map (resp. gaze position) over the training set. For the second, we shuffle ground-truth attention map samples within each lap of the race track in the test set, thus retaining the same overall distribution across that lap but disconnecting the attention output from the RGB input. Furthermore, we compare our results with a state-of-the-art model [48, 49], which also predicts attention maps from single RGB images. As metrics for visual attention prediction, we use the Kullback-Leibler divergence ($D_{KL}$), also used for training our model, and the Pearson Correlation Coefficient ($CC$). The results are shown in Table 1. Our visual attention prediction model (ResNet-18) outperforms the

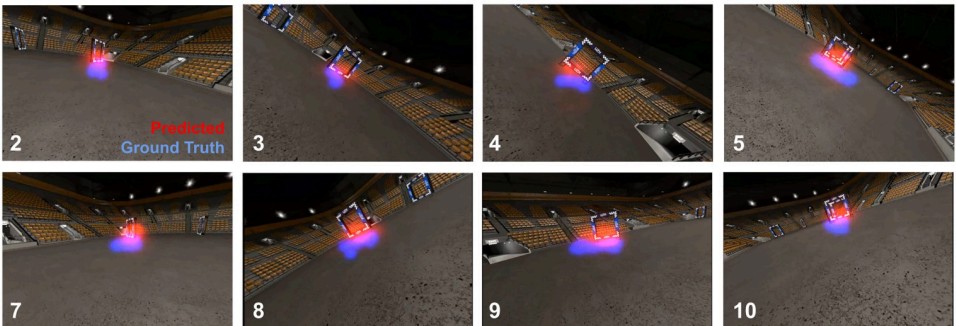

**Fig 4. Visual attention prediction examples.** Comparison of gaze-based attention maps (ground truth, in blue) and visual attention network predictions (in red) for FPV camera images of the left turn maneuver (showing gates 2-5, top row) and the right turn maneuver (showing gates 7-10, bottom row).

**Table 1. Visual attention prediction performance.**

|  | $D_{KL} \downarrow$ | $CC \uparrow$ |
|---|---|---|
| **Baseline mean** | 2.499 | 0.281 |
| **Baseline shuffled GT** | $\infty$ | 0.203 |
| **Deep supervision** | **1.600** | **0.500** |
| **ResNet-18** (ours) | 1.716 | 0.487 |

respective baselines in every metric. Although our model does not outperform the state-of-the-art deep supervision model, it achieves performance close to [49] on our dataset while being faster to train and faster during inference. Our model and [48] are more comparable in terms of training and inference time.

## Control command prediction

We analyze the prediction performance of end-to-end controllers using an offline evaluation method. Specifically, we compare the control commands generated by the neural networks to control commands produced by an MPC controller (which has access to the ground truth quadrotor state), while the MPC controls the quadrotor along 18 reference trajectories on which the networks were previously trained (training set) and hold-out trajectories the networks have not previously observed (test set). We use as performance metrics the Mean Squared Error (*MSE*) and Mean Absolute Error (*L*1) for each control command (i.e., Throttle, Roll, Pitch, Yaw) computed across the respective datasets. Table 2 shows results of the control command prediction analysis on the training set. The attention-prediction based controller produces control commands that more closely resemble control commands of the MPC as

**Table 2. Training set control command prediction errors for end-to-end controllers.**

|  | Throttle | | Roll | | Pitch | | Yaw | |
|---|---|---|---|---|---|---|---|---|
|  | MSE $\downarrow$ | L1 $\downarrow$ | MSE $\downarrow$ | L1 $\downarrow$ | MSE $\downarrow$ | L1 $\downarrow$ | MSE $\downarrow$ | L1 $\downarrow$ |
| **RGB images** | 0.51 | 0.58 | 0.59 | 0.69 | 0.60 | 0.60 | 0.14 | 0.31 |
| **Feature tracks** | 0.62 | 0.63 | **0.20** | **0.36** | 0.58 | 0.57 | 0.04 | 0.15 |
| **Attention prediction** (ours) | **0.45** | **0.54** | 0.23 | 0.39 | **0.51** | **0.56** | **0.04** | **0.14** |

**Table 3. Test set control command prediction errors for end-to-end controllers.**

| | Throttle | | Roll | | Pitch | | Yaw | |
|---|---|---|---|---|---|---|---|---|
| | MSE ↓ | L1 ↓ | MSE ↓ | L1 ↓ | MSE ↓ | L1 ↓ | MSE ↓ | L1 ↓ |
| **RGB images** | 1.21 | **0.81** | 5.35 | 0.68 | 1.76 | 0.67 | 543.10 | 0.76 |
| **Feature tracks** | **1.17** | 0.85 | **0.29** | **0.42** | **0.90** | **0.71** | **0.07** | **0.20** |
| **Attention prediction** (ours) | 1.21 | 0.86 | 0.31 | 0.43 | 0.96 | 0.75 | 0.10 | 0.21 |

compared to the image- and feature track-based controller. This indicates that the attention-based controller selects the appropriate control commands more frequently than the image- and feature track-based baselines when deployed on reference trajectories that the controller was trained on.

Table 3 shows the control command prediction performance on the test set. The feature track-based controller shows an overall better match to MPC commands as compared to the attention- and image-based controllers. Thus, the feature-tracks based controller appears to generalize better to previously unseen reference trajectories than the attention- and image-based controllers.

## Drone racing performance

Fig 5 shows a comparison of drone racing performance for the attention-prediction, feature tracks, and image-based end-to-end controllers across 180 trials (i.e., 18 trajectories each flown 10 times) on training set reference trajectories. The attention-prediction based controller successfully completes 159/180 trials (88% success rate) and outperforms both image-based (110/180 trials, 61% success rate) and feature-track based (99/180 trials, 55% success rate) end-to-end controllers.

In Fig 6 we present an analysis of the generalization performance of the chosen end-to-end controllers when attempting to fly reference trajectories of the test set, which none of the networks has previously observed. The attention-prediction based controller again achieves the

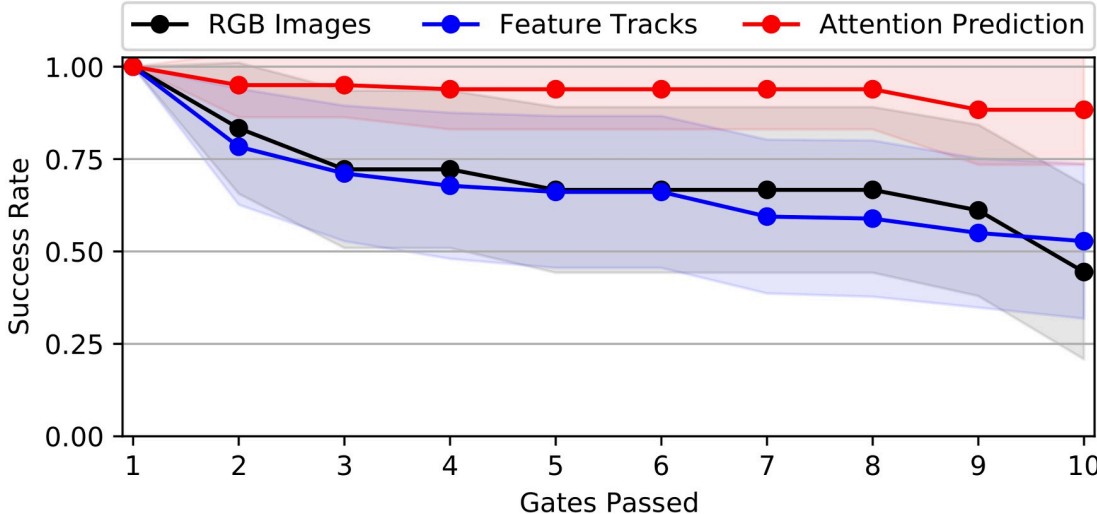

**Fig 5. Training set drone racing performance.** Training set drone racing performance for different end-to-end controllers showing success rates for passing the 10 consecutive gates of the race track. Average success rate and 95% confidence intervals across 18 flight trajectories are shown.

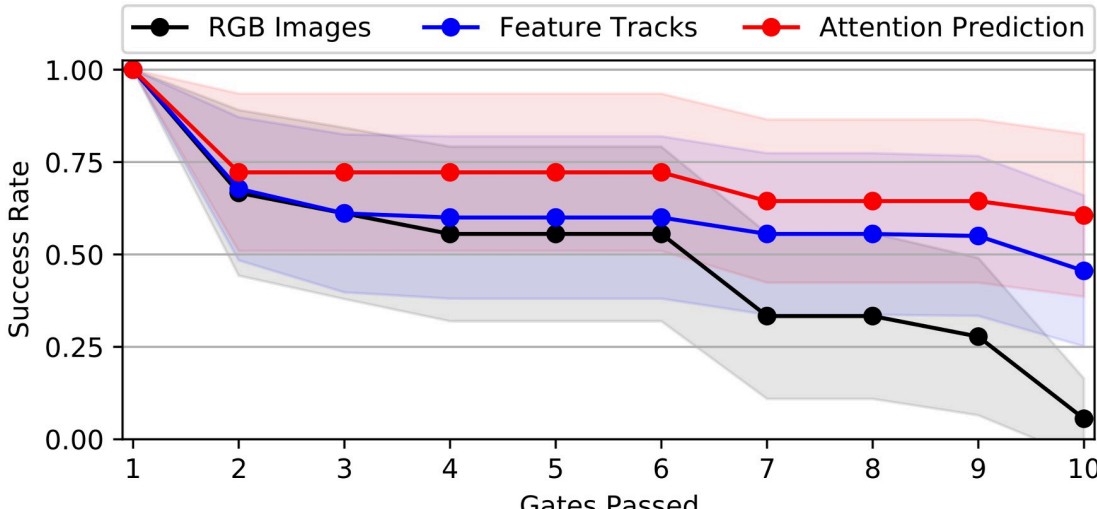

**Fig 6. Test set drone racing performance.** Test set drone racing performance for different end-to-end controllers showing success rates for passing the 10 consecutive gates of the race track. Average success rate and 95% confidence intervals across 18 flight trajectories are shown.

highest number of successfully completed trials (130/180 trials, 72% success rate) and outperforms the feature tracks-based (104/180 trials, 58% success rate) and image-based (70/180 trials, 39% success rate) end-to-end controllers. When comparing controller performance between training and test set, it can be noted that the image-based controller showed a much larger decrease in performance (-22% success rate difference) than the attention-prediction based controller (i.e., -16% success rate difference). The feature-track based controller did not considerably change performance (+3% success rate difference) between training and test set, indicating that the feature-track based controller showed better generalization to previously unseen reference trajectories.

## Discussion

This study investigates whether visual attention prediction can improve the drone racing performance of end-to-end neural network controllers. Our results show that using human drone pilots' eye gaze data we can train a neural network that reliably predicts visual attention when no human is controlling an FPV racing drone. Using this attention prediction network, we successfully train end-to-end neural networks that can fly a challenging race track fully autonomously and collision-free with up to 88% success rate across 180 attempted flight. This attention-prediction based model outperforms controllers based on raw images and feature tracks. Several reasons may contribute to the superior performance of the attention-prediction based controller over the RGB-image and feature-track based controllers. First, attention prediction serves as a task-specific abstraction of image information. That is, attention prediction emulates the eye gaze behavior of human pilots in a drone race, which depends on the pilot's intention ("Pass the next gate") and planned flight trajectory [2]. Indeed, eye gaze has been successfully used as a high-level control input for teleoperated quadrotor navigation [50, 51]. Second, the attention-prediction model may provide useful information for quadrotor state estimation. The attention prediction feature maps typically highlight subregions of the image where the upcoming race gate is located (Fig 1c). This drone-racing specific selection of spatial regions of interest is not available from feature tracks or RGB images alone. Indeed, previous

work has demonstrated that attention prediction models can improve the performance of simultaneous localization and mapping algorithms [52]. Third, attention-prediction and feature-track models reduce the number of input features per sample to the end-to-end controller network (attention prediction: $25 \times 19$ features, feature tracks: $40 \times 5$ features) when compared to raw RGB images ($400 \times 300 \times 3$ features). Our results go beyond the state of the art by showing, for the first time, a successful behavior cloning of human eye-gaze based visual attention and flight behavior of experienced drone racing pilots, achieving human-level, fully autonomous vision-based quadrotor flight. Our work differs from previous model-based and learning-based approaches to autonomous drone racing in the following ways: We do not explicitly encode racing gate poses or relative locations (e.g., as in [15]) but let the attention-prediction model select relevant task visual-spatial information from RGB images. Moreover, by using multiple reference trajectories in training the learned end-to-end controllers, we demonstrate that our controllers can complete multiple reference trajectories despite large variations between the provided reference trajectories. Furthermore, we extend previous works using feature tracks for visual abstraction (e.g., [11]) by showing that visual attention prediction can provide similar and even better performance in vision-based racing tasks. We interpret this result as follows: The visual attention prediction model learns to select task-relevant image features (i.e., vicinity to race gates) that are important for the drone racing task—as shown empirically by [2]. Thus, attention prediction models convey intentionality, which is not provided by purely image feature-based abstractions as provided by feature tracks. This perceptual intentionality can be highly beneficial if the race track and desired trajectory is previously known (i.e., as shown in our drone racing performance analysis on the training set). Nevertheless, feature tracks provide very robust performance on hold-out data, in line with previous observations [11]. Our results extend previous work on gaze-based attention prediction originally carried out for autonomous driving [32, 33] to fast and agile quadrotor flight in three dimensions. One may ask whether the attention-prediction based end-to-end controller could be deployed on a quadrotor platform flying in the real world? We think that real-world deployment is feasible because in our previous work [11] a feature-track-based end-to-end controller was successfully deployed on an NVIDIA Jetson TX2 for acrobatic flight in the real world. Furthermore, in our present work, both the feature-track and attention-prediction-based controllers successfully performed output predictions within 40 ms sample-to-sample intervals. However, further work will be needed to evaluate simulation-to-reality transfer for the attention-prediction model. Potential future applications of human-attention based autonomous flight are precision agriculture [53], road traffic surveillance [54], internet of things [55, 56], assistive technologies for hands-free remote control [50, 57], inspection [58, 59], and search-and-rescue [60, 61].

## Conclusion

This paper addresses the problem of learning fast and agile quadrotor flight from expert human drone pilots. We consider the question of whether human visual attention prediction can improve the performance of autonomous drone racing agents over state-of-the-art methods. To address the problem of a lack of human ground truth data during autonomous flight, we train a neural network that predicts gaze-based visual attention from RGB images. We systematically compare the performance of end-to-end neural network controllers in an autonomous drone racing task. Our results show that gaze-based visual attention prediction outperformed image-based and feature-tracks based controllers. These results provide an essential step towards human-inspired fully autonomous learning-based vision-based fast and agile flight.

## Acknowledgments

We thank Yunlong Song for help with the Flightmare simulator configuration.

## Author Contributions

**Conceptualization:** Christian Pfeiffer, Antonio Loquercio, Davide Scaramuzza.

**Data curation:** Christian Pfeiffer, Simon Wengeler.

**Formal analysis:** Christian Pfeiffer, Simon Wengeler.

**Funding acquisition:** Christian Pfeiffer, Davide Scaramuzza.

**Investigation:** Christian Pfeiffer, Simon Wengeler.

**Methodology:** Christian Pfeiffer, Simon Wengeler, Antonio Loquercio.

**Project administration:** Christian Pfeiffer.

**Software:** Christian Pfeiffer, Simon Wengeler.

**Supervision:** Christian Pfeiffer, Davide Scaramuzza.

**Validation:** Simon Wengeler.

**Visualization:** Christian Pfeiffer, Simon Wengeler.

**Writing – original draft:** Christian Pfeiffer, Simon Wengeler, Antonio Loquercio, Davide Scaramuzza.

**Writing – review & editing:** Christian Pfeiffer, Simon Wengeler, Antonio Loquercio, Davide Scaramuzza.

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
