## [Decision Letter · Decision Letter 0]

25 Jan 2022

PONE-D-21-40252Visual attention prediction improves performance of autonomous drone racing agentsPLOS ONE

Dear Dr. Pfeiffer,

Thank you for submitting your manuscript to PLOS ONE. After careful consideration, we feel that it has merit but does not fully meet PLOS ONE’s publication criteria as it currently stands. Therefore, we invite you to submit a revised version of the manuscript that addresses the points raised during the review process.

Thank you for submitting your manuscript to PLOS ONE. After careful consideration, we feel that it has merit but does not fully meet PLOS ONE’s publication criteria as it currently stands. Therefore, we invite you to submit a revised version of the manuscript that addresses the points raised during the review process.

We look forward to receiving your revised manuscript.

Kind regards,

Sathishkumar V E

Academic Editor

PLOS ONE

Journal Requirements:

"This work was supported by the Ernst Göhner Foundation and University of Zurich Alumni Fonds zur Förderung des Akademischen Nachwuchses (FAN Fellowship) to C.P., by the National Centre of Competence in Research (NCCR) Robotics through the Swiss National Science Foundation (SNSF) to and the European Union’s Horizon 2020 Research and Innovation Programme under grant agreement No. 871479 (AERIAL-CORE) and the European Research Council (ERC) under grant agreement No. 864042 (AGILEFLIGHT) to D.S.."

Reviewers' comments:

Reviewer's Responses to Questions

**Comments to the Author**

1. Is the manuscript technically sound, and do the data support the conclusions?

Reviewer #1: Yes

Reviewer #2: Partly

2. Has the statistical analysis been performed appropriately and rigorously? 

Reviewer #1: Yes

Reviewer #2: No

3. Have the authors made all data underlying the findings in their manuscript fully available?

Reviewer #1: Yes

Reviewer #2: No

4. Is the manuscript presented in an intelligible fashion and written in standard English?

Reviewer #1: Yes

Reviewer #2: No

5. Review Comments to the Author

Reviewer #1: This paper study demonstrates that gaze-based visual attention prediction can improve the performance of end-to-end controllers for autonomous drone racing, providing an essential step towards human-inspired fully autonomous learning-based vision-based fast and agile flight. This article is novel and the contributions are suitable for a journal article. However, the following minor comments to be addressed before consider this for publication in this journal.

1. Provide the citations for the datasets.

2. What are the primary reason to achieve the superior performance of the proposed work over the existing models

3. List the contributions in the introduction.

4. Add a separate section for the literature study.

Reviewer #2: The Paper needs the following Major Revisions and is subject for re-review, and after re-review the final decision for the paper will be done:

1. Abstract- Highlight in what %age and in what parameters the proposed methodology is found better and as compared to existing techniques and what is the overall analysis of the proposed technique.

2. Introduction should mention more information with regard to the scope and problem definition. And add the Objectives of the paper at end of Introduction. Add organization of the paper.

3. Literature review is missing in the paper. It is suggested to add min 15-25 papers to the paper, and every paper should be highlighted with what is being proposed, what is the novelty and what experimental results are observed. It is also suggested to add 9-15 lines at the end of Literature review, which highlights the overall technical gaps of the paper.

4. Add the section with regard to the proposed methodology and should be elaborated with System Model and other technical highlights of the proposed model.

5. Add some Real time case study based discussion to the paper.

6. Conclusion should be more broad and should focus on the proposed work, experimental results and even on the performance comparison. Addition of future scope is suggested to the paper.

7. Add the following references to the paper:

a. Puri, V., Nayyar, A., & Raja, L. (2017). Agriculture drones: A modern breakthrough in precision agriculture. Journal of Statistics and Management Systems, 20(4), 507-518.

b. Nayyar, A., Nguyen, B. L., & Nguyen, N. G. (2020). The internet of drone things (IoDT): future envision of smart drones. In First international conference on sustainable technologies for computational intelligence (pp. 563-580). Springer, Singapore.

c. Kumar, A., Krishnamurthi, R., Nayyar, A., Luhach, A. K., Khan, M. S., & Singh, A. (2021). A novel Software-Defined Drone Network (SDDN)-based collision avoidance strategies for on-road traffic monitoring and management. Vehicular Communications, 28, 100313.

d. Khan, N. A., Jhanjhi, N. Z., Brohi, S. N., & Nayyar, A. (2020). Emerging use of UAV’s: secure communication protocol issues and challenges. In Drones in Smart-Cities (pp. 37-55). Elsevier.

6. PLOS authors have the option to publish the peer review history of their article (what does this mean?). If published, this will include your full peer review and any attached files.

Reviewer #1: No

Reviewer #2: No

---

## [Author Response · Author response to Decision Letter 0]

9 Feb 2022

Response to Reviewers

Dear Editor and Reviewers,

We would like to thank you for the positive feedback and for considering our paper for publication. We have revised the manuscript according to the comments of the Editor and Reviewers and highlighted all changes to the manuscript in blue. Below we present our detailed responses to the Editor and Reviewers in blue.

Sincerely yours,

Christian Pfeiffer

Simon Wengeler

Antonio Loquercio

Davide Scaramuzza

Editor

Editor Comment 1: Thank you for stating the following financial disclosure: "This work was supported by the Ernst Göhner Foundation and University of Zurich Alumni Fonds zur Förderung des Akademischen Nachwuchses (FAN Fellowship) to C.P., by the National Centre of Competence in Research (NCCR) Robotics through the Swiss National Science Foundation (SNSF) to and the European Union’s Horizon 2020 Research and Innovation Programme under grant agreement No. 871479 (AERIAL-CORE) and the European Research Council (ERC) under grant agreement No. 864042 (AGILEFLIGHT) to D.S.." Please state what role the funders took in the study. If the funders had no role, please state: "The funders had no role in study design, data collection and analysis, decision to publish, or preparation of the manuscript." 

If this statement is not correct you must amend it as needed. Please include this amended Role of Funder statement in your cover letter; we will change the online submission form on your behalf.

Response to Editor Comment 1: We added an updated financial disclosure statement to the Cover Letter: “This work was supported by the Ernst Göhner Foundation and University of Zurich Alumni Fonds zur Förderung des Akademischen Nachwuchses (FAN Fellowship), by the National Centre of Competence in Research (NCCR) Robotics through the Swiss National Science Foundation (SNSF) and the European Union’s Horizon 2020 Research and Innovation Programme under grant agreement No. 871479 (AERIAL-CORE) and the European Research Council (ERC) under grant agreement No. 864042 (AGILEFLIGHT). The funders had no role in study design, data collection and analysis, decision to publish, or preparation of the manuscript.”.

Editor Comment 2: We note that you have stated that you will provide repository information for your data at acceptance. Should your manuscript be accepted for publication, we will hold it until you provide the relevant accession numbers or DOIs necessary to access your data. If you wish to make changes to your Data Availability statement, please describe these changes in your cover letter and we will update your Data Availability statement to reflect the information you provide.

Response to Editor Comment 2: We added the DOI number for accessing the dataset to the Cover Letter and the revised Methods section (lines 184-185): “The dataset used in this study is available in an Open Science Framework repository (https://osf.io/uabx4/, Dataset DOI: 10.17605/OSF.IO/UABX4).”.

Reviewer #1

This paper study demonstrates that gaze-based visual attention prediction can improve the performance of end-to-end controllers for autonomous drone racing, providing an essential step towards human-inspired fully autonomous learning-based vision-based fast and agile flight. This article is novel and the contributions are suitable for a journal article. However, the following minor comments to be addressed before considering this for publication in this journal.

Reviewer #1 Comment 1: Provide the citations for the datasets.

Response to Reviewer #1 Comment 1: We added the DOI number for accessing the dataset to the revised Methods section (lines 184-185): “The dataset used in this study is available in an Open Science Framework repository (https://osf.io/uabx4/, Dataset DOI: 10.17605/OSF.IO/UABX4).”.

Reviewer #1 Comment 2: What are the primary reason to achieve the superior performance of the proposed work over the existing models

Response to Reviewer #1 Comment 2: We added a discussion of this point to the revised Discussion section (lines 326-341): “Several reasons may contribute to the superior performance of the attention-prediction based controller over the RGB-image and feature-track based controllers. First, attention prediction serves as a task-specific abstraction of image information. That is, attention prediction emulates the eye gaze behavior of human pilots in a drone race, which depends on the pilot’s intention (“Pass the next gate”) and planned flight trajectory [2]. Indeed, eye gaze has been successfully used as a high-level control input for teleoperated quadrotor navigation [50, 51]. Second, the attention-prediction model may provide useful information for quadrotor state estimation. The attention prediction feature maps typically highlight subregions of the image where the upcoming race gate is located (Fig. 1c). This drone-racing specific selection of spatial regions of interest is not available from feature tracks or RGB images alone. Indeed, previous work has demonstrated that attention prediction models can improve the performance of simultaneous localization and mapping algorithms [52]. Third, attention-prediction and feature-track models reduce the number of input features per sample to the end-to-end controller network (attention prediction: 25×19 features, feature tracks: 40×5 features) when compared to raw RGB images (400×300×3 features).”.

Reviewer #1 Comment 3: List the contributions in the introduction.

Response to Reviewer #1 Comment 3: We report the contributions in the revised Contributions section (lines 73-83): “The main contributions of this work are: 

We train and evaluate a visual attention prediction model for autonomous drone racing.

We train end-to-end deep learning networks using imitation learning that can complete a challenging race in a vision-based drone racing task, with a performance as good as human pilots. 

We demonstrate that attention prediction models outperform models using raw image inputs and image-based abstractions (i.e., feature tracks).

We found a better generalization performance to previously unseen flight trajectories for end-to-end drone racing agents using attention prediction or feature tracks when compared to a raw image input baseline.”.

Reviewer #1 Comment 4: Add a separate section for the literature study.

Response to Reviewer #1 Comment 4: We moved the literature review to the revised Related Work section (lines 90-137).

Reviewer #2

The Paper needs the following Major Revisions and is subject for re-review, and after re-review the final decision for the paper will be done:

Reviewer #2 Comment 1: Abstract- Highlight in what %age and in what parameters the proposed methodology is found better and as compared to existing techniques and what is the overall analysis of the proposed technique.

Response to Reviewer #2 Comment 1: We added this information to the revised Abstract (lines 21-23): “Comparing success rates for completing a challenging race track by autonomous flight, our results show that the attention-prediction based controller (88% success rate) outperforms the RGB-image (61% success rate) and feature-tracks (55% success rate) controller baselines.”.

Reviewer #2 Comment 2: Introduction should mention more information with regard to the scope and problem definition. And add the Objectives of the paper at end of Introduction. Add organization of the paper.

Response to Reviewer #2 Comment 2: We added the objective, scope, and problem definition to the revised Introduction section (lines 64-71): “The main objective of this work is to answer the question of whether gaze-based visual attention prediction can improve the performance of end-to-end models for vision-based autonomous drone racing beyond state-of-the-art. We address the problem of a lack of human ground truth data during deployment by training a neural network for predicting human visual attention from RGB images. The scope of the present work is an evaluation of the flight performances of end-to-end controller architectures for the task of vision-based autonomous drone racing in a highly realistic simulator.”. In addition, we describe the organization of the paper in the revised Contributions section (lines 84-89): “The Related Work section describes related works in the domain. The Materials and Methods section describes the datasets, network architectures, and experimental analysis methods used in this work. The Results section presents experimental results obtained for the visual attention prediction, control command prediction, and end-to-end drone racing performance. The Discussion section relates the experimental findings to previous work and proposed future work. The Conclusion section concludes the paper.”.

Reviewer #2 Comment 3: Literature review is missing in the paper. It is suggested to add min 15-25 papers to the paper, and every paper should be highlighted with what is being proposed, what is the novelty and what experimental results are observed. It is also suggested to add 9-15 lines at the end of Literature review, which highlights the overall technical gaps of the paper.

Response to Reviewer #2 Comment 3: We extended the literature review in the revised Related Work section (lines 102-120): “Another shortcoming of imitation learning is that it does not allow the network to compensate for mistakes made by the expert. A possible solution is the use of observational imitation learning in which a network learns to select optimal behavior while observing multiple imperfect teachers. This approach outperformed reinforcement learning and imitation learning approaches in vision-based autonomous drone racing in a simulator [24]. However, not only the choice of network architecture and training method but also the choice of input/output representation strongly affect network performance. Abstractions of either input or output data typically outperform networks operating directly on raw image data. For instance, [11] observed better performance in autonomous acrobatic flight using feature tracks than using RGB images directly. Similarly, [25] found better 3D localization performance using grayscale instead of RGB images. Likewise, [26] found better performances in autonomous car racing when predicting parameterized trajectories for a model predictive controller (MPC) driving the car compared to letting the network predict control commands directly. Such sensory and output abstractions seem advantageous in network performance and generalization ability. It should also be noted that several previous works follow hybrid approaches combining learning methods for perception [27] and localization [28] with model-based methods for planning [29] and control [21] and have demonstrated successes. However, these approaches often require extensive system identification and controller tuning, which are not required when using end-to-end neural network controllers.”.

Reviewer #2 Comment 4: Add the section with regard to the proposed methodology and should be elaborated with System Model and other technical highlights of the proposed model.

Response to Reviewer #2 Comment 4: We added information about the Quadrotor platform to the revised Methods section (lines 159-162): “The quadrotor platform had an arm length of 17 cm, an all-up-weight of 1 kg, a maximum collective thrust of 21.7 N, and a maximum rotational velocity of 6 rad/s. The RGB camera had a horizontal field-of-view of 80°, and an uptilt angle of 25°.”.

Reviewer #2 Comment 5: Add some Real time case study based discussion to the paper.

Response to Reviewer #2 Comment 5: We are not completely sure what the Reviewer means. We provide answers to two interpretations of the Reviewer’s question. Question 1: Would your method work on a real quadrotor platform in real-time? Answer 1: This question cannot be directly answered from our study results conducted in the Flightmare simulator. For context, we use a simplified simulation loop that executes the physics simulation, video rendering, and control command generation sequentially, rather than concurrently. Processing of the visual inputs, attention map prediction, feature track extraction/tracking, and network command generation are performed at fixed pre-defined time intervals rather than being dependent on the hardware performance for each of these tasks. The simulation, therefore, does not have a direct relation to the possible real-time performance of a system using our proposed models in a more realistic setting. However, in our previous work, we successfully deployed an end-to-end neural network using feature-track inputs on an NVIDIA Jetson TX2 and successfully demonstrated agile flight maneuvers (Kaufmann et al. 2020). Given that we did not observe a notable performance difference between the feature-track and attention-prediction-based controllers in terms of inference time, we would expect that real-world deployment of our end-to-end controllers is feasible. Question 2: What is the inference time of the networks? Answer 2: Running the simulation using network-generated commands resulted in close to real-time execution for all architectures on a machine with an Nvidia GTX 970 graphics card and an Intel Core i5-4690k CPU. This suggests that all controller architectures are able to perform inference in at most 40 ms (based on command generation at 25 Hz). In all our experiments we sample output commands at 25 Hz. All networks thus had 40 ms time for inference. We address these points in the revised Discussion (lines 359-366): “One may ask whether the attention-prediction based end-to-end controller could be deployed on a quadrotor platform flying in the real world? We think that real-world deployment is feasible because in our previous work [11] a feature-track-based end-to-end controller was successfully deployed on an NVIDIA Jetson TX2 for acrobatic flight in the real world. Furthermore, in our present work, both the feature-track and attention-prediction-based controllers successfully performed output predictions within 40 ms sample-to-sample intervals. However, further work will be needed to evaluate simulation-to-reality transfer for the attention-prediction model.”.

Reviewer #2 Comment 6: Conclusion should be more broad and should focus on the proposed work, experimental results and even on the performance comparison. Addition of future scope is suggested to the paper.

Response to Reviewer #2 Comment 6: We extended the revised Conclusion section (lines 371-379): “This paper addresses the problem of learning fast and agile quadrotor flight from expert human drone pilots. We consider the question of whether human visual attention prediction can improve the performance of autonomous drone racing agents over state-of-the-art methods. To address the problem of a lack of human ground truth data during autonomous flight, we train a neural network that predicts gaze-based visual attention from RGB images. We systematically compare the performance of end-to-end neural network controllers in an autonomous drone racing task. Our results show that gaze-based visual attention prediction outperformed image-based and feature-tracks based controllers. These results provide an essential step towards human-inspired fully autonomous learning-based vision-based fast and agile flight.”.

Reviewer #2 Comment 7: Add the following references to the paper:

a. Puri, V., Nayyar, A., & Raja, L. (2017). Agriculture drones: A modern breakthrough in precision agriculture. Journal of Statistics and Management Systems, 20(4), 507-518.

b. Nayyar, A., Nguyen, B. L., & Nguyen, N. G. (2020). The internet of drone things (IoDT): future envision of smart drones. In First international conference on sustainable technologies for computational intelligence (pp. 563-580). Springer, Singapore.

c. Kumar, A., Krishnamurthi, R., Nayyar, A., Luhach, A. K., Khan, M. S., & Singh, A. (2021). A novel Software-Defined Drone Network (SDDN)-based collision avoidance strategies for on-road traffic monitoring and management. Vehicular Communications, 28, 100313.

d. Khan, N. A., Jhanjhi, N. Z., Brohi, S. N., & Nayyar, A. (2020). Emerging use of UAV’s: secure communication protocol issues and challenges. In Drones in Smart-Cities (pp. 37-55). Elsevier.

Response to Reviewer #2 Comment 7: We added the citations to the revised Discussion section (lines 346-348): “Potential future applications of human-attention based autonomous flight are precision agriculture [53], road traffic surveillance [54], internet of things [55, 56], assistive technologies for hands-free remote control [50, 57], inspection [58, 59], and search-and-rescue [60, 61].”.

---

## [Decision Letter · Decision Letter 1]

11 Feb 2022

Visual attention prediction improves performance of autonomous drone racing agents

PONE-D-21-40252R1

Dear Dr. Pfeiffer,

We’re pleased to inform you that your manuscript has been judged scientifically suitable for publication and will be formally accepted for publication once it meets all outstanding technical requirements.

Kind regards,

Sathishkumar V E

Academic Editor

PLOS ONE

Additional Editor Comments (optional):

Reviewers' comments:

Reviewer's Responses to Questions

**Comments to the Author**

1. If the authors have adequately addressed your comments raised in a previous round of review and you feel that this manuscript is now acceptable for publication, you may indicate that here to bypass the “Comments to the Author” section, enter your conflict of interest statement in the “Confidential to Editor” section, and submit your "Accept" recommendation.

Reviewer #1: All comments have been addressed

Reviewer #2: All comments have been addressed

2. Is the manuscript technically sound, and do the data support the conclusions?

Reviewer #1: Yes

Reviewer #2: Yes

3. Has the statistical analysis been performed appropriately and rigorously? 

Reviewer #1: Yes

Reviewer #2: Yes

4. Have the authors made all data underlying the findings in their manuscript fully available?

Reviewer #1: Yes

Reviewer #2: Yes

5. Is the manuscript presented in an intelligible fashion and written in standard English?

Reviewer #1: Yes

Reviewer #2: Yes

6. Review Comments to the Author

Reviewer #1: The authors addressed all the comments and the current version is well improved. So this version is recommended for publication in this journal. Congratulations to the authors.

Reviewer #2: The revised paper has incorporated all the revisions as suggested in the last review. ANd now the paper stands Accepted with no further revisions.

7. PLOS authors have the option to publish the peer review history of their article (what does this mean?). If published, this will include your full peer review and any attached files.

Reviewer #1: No

Reviewer #2: No

---

## [Editor Report · Acceptance letter]

21 Feb 2022

PONE-D-21-40252R1 

Visual attention prediction improves performance of autonomous drone racing agents 

Dear Dr. Pfeiffer:

I'm pleased to inform you that your manuscript has been deemed suitable for publication in PLOS ONE. Congratulations! Your manuscript is now with our production department. 

Kind regards, 

on behalf of

Dr. Sathishkumar V E 

Academic Editor

PLOS ONE